# Identification of Photosynthesis Characteristics and Chlorophyll Metabolism in Leaves of *Citrus* Cultivar (*Harumi*) with Varying Degrees of Chlorosis

**DOI:** 10.3390/ijms24098394

**Published:** 2023-05-07

**Authors:** Bo Xiong, Ling Li, Qin Li, Huiqiong Mao, Lixinyi Wang, Yuhui Bie, Xin Zeng, Ling Liao, Xun Wang, Honghong Deng, Mingfei Zhang, Guochao Sun, Zhihui Wang

**Affiliations:** 1College of Horticulture, Sichuan Agricultural University, Chengdu 611130, China; 2Institute of Pomology and Olericulture, Sichuan Agricultural University, Chengdu 611130, China

**Keywords:** *Citrus*, chlorosis, photosynthetic characteristics, chlorophyll synthesis, chlorophyll degradation

## Abstract

In autumn and spring, citrus leaves with a *Ponkan* (*Citrus reticulata* Blanco cv. Ponkan) genetic background (*Harumi*, *Daya*, etc.) are prone to abnormal physiological chlorosis. The effects of different degrees of chlorosis (normal, mild, moderate and severe) on photosynthesis and the chlorophyll metabolism of leaves of *Citrus* cultivar (*Harumi*) were studied via field experiment. Compared with severe chlorotic leaves, the results showed that chlorosis could break leaf metabolism balance, including reduced chlorophyll content, photosynthetic parameters, antioxidant enzyme activity and enzyme activity related to chlorophyll synthesis, increased catalase and decreased enzyme activity. In addition, the content of chlorophyll synthesis precursors showed an overall downward trend expected for uroporphyrinogen III. Furthermore, the relative expression of genes for chlorophyll synthesis (*HEMA1*, *HEME2*, *HEMG1* and *CHLH*) was down-regulated to some extent and chlorophyll degradation (*CAO*, *CLH*, *PPH*, *PAO* and *SGR*) showed the opposite trend with increased chlorosis. Changes in degradation were more significant. In general, the chlorosis of *Harumi* leaves might be related to the blocked transformation of uroporphyrinogen III (Urogen III) to coproporphyrinogen III (Coprogen III), the weakening of antioxidant enzyme system activity, the weakening of chlorophyll synthesis and the enhancement in degradation.

## 1. Introduction

*Citrus*, one of the world’s most important fruit crops in the family Rutaceae [1], is grown in more than 140 countries, mainly in tropical and subtropical regions [2]. Citrus fruits are rich in bioactive substances, such as essential oils, carotenoids, flavonoids, acacia and limonoids, which have significant effects in terms of anti-inflammation, anti-oxidation, immune regulation, prevention and treatment of multiple respiratory diseases [3,4]. *Harumi*, which is a late-maturing hybrid citrus [‘F 2432’ *Ponkan* (*Citrus reticulata*) × *Kiyomi* tangor (*Citrus unshiu* × *Citrus sinensis*)] [5], has been increasingly planted in China and favored by many consumers because of its high quality, high yield and abundant nutrients [6]. However, after years of field observation, we found that citrus (*Harumi*, *Daya*, etc.) leaves with a *Ponkan* (*Citrus reticulata* Blanco cv. *Ponkan*) genetic background were prone to abnormal physiological chlorosis. Leaf color variation is a common type of mutation, which is usually caused by abnormal chlorophyll synthesis and metabolism pathways caused by gene mutation [7]. The most common leaf color mutations are chlorosis and albino. Chlorotic leaf is a product of reduced chlorophyll content. Many scholars have explored the cause and found that disrupted chlorophyll biosynthesis [8], accelerated chlorophyll degradation [9] and reduced chlorophyll protection activity and pigment [10] may lead to changes in plant leaf color.

Chlorophyll (Chl) is the main photosynthetic pigment of plants. The proper provision of Chl is a key condition for the smooth performance of photosynthesis [11]. Chlorophylls (Chls, Chl *a* and Chl *b*) are tetrapyrrole molecules essential for photosynthetic light harvesting and energy transduction [12]. Changes in natural conditions, such as light and temperature, affect the process of chlorophyll synthesis and degradation, resulting in changes in Chl content. When Chl content changes in plants, various leaf color mutant phenotypes occur, including chlorina, virescent, albino, yellow-green and stay-green. Leaf color mutants develop from the inhibition of genes that regulate Chl biosynthesis and chloroplast development [13]. The total Chl content per unit of leaf area is one of the most sensitive factors to nutrient availability and different environmental stresses [14]. In addition, chlorophyll *a* (Chl *a*) is essential for photochemistry, which, in its excited state, converts light energy into electricity, while chlorophyll *b* (Chl *b*) provides plants with an advantage in harvesting light around 450 nm, a wavelength region of light that is not efficiently absorbed by Chl *a* [11,15]. However, Chl, which is a potential cellular phytotoxin, must be degraded when there is excess Chl and its derivatives [16]. Otherwise, reactive oxygen species (ROS) produced by excess Chl can lead to leaf cell death when Chl degradation is inhibited [17,18].

In recent years, the study of chlorophyll synthesis and the degradation pathway has become a hot topic [19,20,21]. Chl metabolism is strictly regulated at different stages of plant development [14]. A study on tomato chlorophyll has shown that the *slym1* gene negatively regulates photosynthesis. The binding of *Slym1* to the intermediate protein MP (ID: 101256896) leads to the interaction between *MP* and *HY5*, resulting in the accumulation of a large amount of unbound *HY5* in the cell, thus accelerating the breakdown of chlorophyll [22]. Chlide a oxygenase (CAO) is the only enzyme found to catalyze the conversion of Chl *a* to Chl *b* [12]. *SGR1* plays a key role in chlorophyll degradation, and plants lacking *SGR1* exhibit a strong green-preserving phenotype [23,24]. Pentatricopeptide repeat (PPR) proteins have been shown to influence chloroplast development and chlorophyll accumulation by editing the RNA of chloroplast genes [25]. Soluble versus membrane localization of glutamyl-tRNA reductase (GluTR) has also been shown to be related to chlorophyll content [26]. The Chl biosynthesis pathway in plant tissue is now largely elucidated, but its catabolism regulation is incomplete. Leaf senescence is caused by the breakdown of chlorophyll and the resulting chlorosis of the leaves. In these processes, chlorophyll is broken down in the PAO/Phyllobilin pathway [27,28].

Many studies have focused on the analysis of the expression of relevant genes during photosynthesis, and the regulation of corresponding protein levels [29,30]. However, in this experiment, we investigated photosynthesis, the antioxidant enzyme system, chlorophyll synthesis and degradation metabolism in the common citrus variety *Harumi*. Chlorophyll metabolism is very complex. Among them, the process of chlorophyll biosynthesis takes L-glutamy-tRNA as a substrate. Under the joint action of a variety of enzymes and related genes, it takes 16 steps to finally synthesize chlorophyll [31]. In chlorophyll synthesis, any problem with any link may lead to abnormal chlorophyll metabolism [25]. Therefore, it is necessary to further study the intermediates, enzyme activities and gene expression related to the chlorophyll metabolism pathway. It is also of great significance to investigate the causes of abnormal chlorosis (one-year-old leaves on autumn shoots, senescence of leaves not yet occurred) of *Harumi* leaves to promote the normal growth, yield and quality formation of citrus fruits.

## 2. Results

### 2.1. Photosynthetic Pigments and Photosynthetic Parameters

Photosynthetic pigments in leaves decreased with the degree of chlorosis (Figure 1B). The photosynthetic pigments of normal leaves were significantly higher than leaves with moderate and severe chlorosis. There was no significant difference in the chlorophyll a/b ratio, but there was a significant difference in the chlorophyll/carotenoid ratio. The chlorophyll/carotenoid ratio decreased significantly with the increase in chlorosis degree, and the maximum decrease was 18.50%. The analysis of gas exchange parameters showed that as chlorosis deepened, Pn (net photosynthesis rate), Tr (transpiration rate) and Gs (stomatal conductance) values significantly decreased, excluding an increase in Ci (intercellular CO_2_ concentration) (Figure 1C). Pn, Tr and Gs showed linear regression with R^2^ = 0.9928, R^2^ = 0.9791 and R^2^ = 0.9776 with total chlorophyll, respectively. However, no linear correlation was found between Ci and total chlorophyll (Table 1).

### 2.2. Chlorophyll Synthesis Precursors

The contents of Coprogen III, Proto IX (protoporphyrin IX), Mg-Proto IX (Mg-protoporphyrin IX) and Pchlide (protochlorophyllide) in chlorotic leaves were significantly lower than those of the normal leaves, except for Urogen III, ALA (δ-5-aminolevulinic acid) and PBG (porphobilinogen) (Figure 2). Changes in ALA and PBG contents were basically the same, decreasing in mild and moderate chlorotic leaves and then increasing in severe chlorotic leaves (Figure 2A). The higher the degree of chlorosis, the lower the content of Coprogen III, Proto IX, Mg-Proto IX and Pchlide, but the higher the content of Urogen III. All of the levels of chlorosis represented significant differences for each other (excluding Pchlide) (Figure 2B). Pchlide content showed a decreasing trend while there were no significant differences between the normal and mild chlorosis leaves (Figure 2C). Compared with the normal leaves, the precursors of severe chlorosis had extremely significant changes. Urogen III, Pchlide, Coprogen III, Mg-Proto IX and Proto IX in leaves with severe chlorosis were 139.4%, 49.9%, 31.6%, 25.6% and 1.6% of normal leaves, respectively.

### 2.3. Activity of Chlorophyll Synthesis and Degradation Enzyme

In chlorotic *Harumi* leaves, the highest levels of Glu-TR (Glu-tRNAs), UROD (uroporphyrinogen decarboxylase) and ChlM (magnesium protoporphyrin IX methyltransferase) activity were detected in normal leaves (Figure 3A), and the lowest levels were found in leaves with severe chlorosis. Activity decreased continuously and significantly throughout the chlorosis process. In addition, UROD activity was negatively correlated with the content of Urogen III (R^2^ = −0.8893), but positively correlated with Croprogen III (R^2^ = 0.9824) (Figure 3A). There were no significant differences in the activities of Chlase (chlorophyllase) and MDCase (Mg-dechelatase) as the degrees of chlorosis increased. In the process of deepening the degree of chlorosis, there was no significant change in the chlase activity in leaves (Figure 3B). For Mg-dechelatase activity, although there was a tendency for it to increase, only a significant difference was observed between severe chlorosis and normal leaves, but no significant difference was found between normal and mild chlorosis or moderate and severe chlorosis (Figure 3C).

### 2.4. Relative Expression of Genes Involved in Chlorophyll Metabolism Pathway

Relative expression data showed that the chlorosis of *Harumi* leaves had significant effects on key genes in the chlorophyll synthesis and degradation pathway chlorosis (Figure 4). As chlorosis intensified, downstream *CHLH* expression for chlorophyll synthesis was significantly down-regulated, while upstream *HEMA1*, *HEME2* and *HEMG1* were also down-regulated to some degree. In addition, *HEMA1* and *HEME2* were expressed the highest in mild chlorotic leaves, which were significantly higher than normal, moderate and severe chlorotic leaves. It could be seen that the chlorosis of *Harumi* leaves may have been due to the blocking of the chlorophyll synthesis pathway.

Key gene data for chlorophyll degradation indicated that the leaf chlorosis of *Harumi* increased the expression of *CLH*, *PPH*, *PAO* and *SGR*, and the expression of *CAO* involved in chlorophyll b (chl *b*) synthesis was significantly down-regulated during leaf chlorosis. The maximum expression of *CLH*, *PPH* and *PAO* was found in mild chlorotic leaves, which was consistent with the expression of the above metabolic genes *HEMA1* and *HEME2*. In addition, stay-green (SGR), which was significantly down-regulated in severe chlorotic leaves, was lower than mild and moderate chlorotic leaves, but still higher than normal leaves. The results showed that the chlorophyll degradation was promoted in chlorotic *Harumi* leaves.

### 2.5. Antioxidant Enzyme Activity

With the exception of CAT (catalase), all antioxidant enzyme activities in the citrus leaves of *Harumi* decreased with the deepening of chlorosis (Figure 5). SOD (superoxide dismutase), POD (peroxidase) and APX (ascorbate peroxidase) enzyme activities decreased with increasing degrees of chlorosis, but CAT activity increased. Moreover, the values showed significant variation in the activity of POD and CAT enzymes. The significant difference only existed between normal and severe chlorotic leaves for SOD. APX activity in different degrees of chlorosis was significantly different from normal leaves (Figure 5D).

### 2.6. Correlation Analysis

According to the analysis, we observed both positive and negative correlations between the indicators (Figure 6). There were significant negative correlations between chlorosis index, photosynthesis and chlorophyll parameters (*p* ≤ 0.05). The expression of *SGR*, *PAO*, *PPH* and *CLH* was positively correlated with the chlorosis index, while *HEMA1*, *HEME2* and *HEMG1* had no significant correlation with the chlorosis index (Figure 6A). Notably, ALA and PBG content were significantly negatively correlated with the expression of *SGR* (*p* ≤ 0.05), but there was no significant correlation with other indicators (Figure 6B). The expression of key genes involved in chlorophyll catabolism (*CLH*, *PPH*, *PAO* and *SGR*) was negatively correlated with PBG, Coprogen III, Proto-IX, Mg-Proto IX and Pchlide content; however, a positive correlation was found between the expression of chlorophyll degradation genes (except *SGR*) and Urogen III content. Moreover, there was a positive correlation between the expression of chlorophyll catabolism genes and CAT activity, Chlase activity and Mg-dechelatase activity. The expression of key genes in chlorophyll catabolism was negatively correlated with the expression of chlorophyll synthesis genes (*HEMA1*, *HEME2*, *HEMG1*, *CHLH* and *CAO*), although not significantly. 

## 3. Discussion

Before this experiment, we found that the soil in the sampling garden lacked Mg by measuring the content of mineral elements, which led to the chlorosis of leaves of *Citrus* cultivar (*Harumi*). Via the determination of other indicators of leaves in this experiment, a more comprehensive and in-depth exploration was conducted of the physiological mechanism of chlorosis of *Harumi* leaves.

As chlorosis increased, Pn, Tr and Gs showed a significant downward trend, while Ci showed a significant upward trend. It can be concluded that the fact that the photosynthetic capacity of leaves decreased and the photosynthetic rate decreased may have been caused by stomatal restriction or nonstomatal restriction factors. Ci decreasing and Ls increasing shows that the photosynthetic rate decreasing mainly comes from the stomatal limitation; Ci increasing and Ls decreasing indicates that nonstomatal limitation is the main cause [32,33]. In this experiment, Ci showed a significant increasing trend, but Ls still needed to be further studied to determine the reason for the decline in the photosynthetic rate in chlorotic leaves. Compared with normal leaves, chlorophyll content, carotenoid content and the chlorophyll/carotenoid ratio of chlorotic leaves showed a significant downward trend. This result indicated that leaf chlorosis was related to the content of the main photosynthetic pigments and the effect of chlorosis on chlorophyll content was greater than that of carotenoids. As the chlorophyll/carotenoid ratio decreased linearly, chlorophyll’s ability to cover the leaf color decreased, and the leaf became chlorotic to varying degrees [34]. Compared to other citrus varieties, chlorophyll b content in normal *Harumi* leaves was very low, which could be attributed to the species characteristics. We hypothesized that this might also have been one of the reasons why citrus leaves with a *Ponkan* (*Citrus reticulata* Blanco cv. Ponkan) genetic background were prone to chlorosis in autumn and spring. In addition, chlorophyll synthesis precursors in chlorotic leaves decreased significantly compared to normal leaves, except for Urogen III. It was speculated that the transformation pathway from Urogen III to coprogen III was blocked, which led to the abnormal blocking of the entire chlorophyll biosynthesis pathway. 

In order to analyze the rules of key substances in chlorophyll metabolism, the activities of enzymes related to chlorophyll metabolism and the expression of related genes were further studied. The results showed that the activities of the chlorophyll synthesis enzymes GluTR, UROD and ChlM decreased significantly with the intensification of chlorosis, but the activities of the chlorophyll degradation enzymes Chlase and MDCase did not change significantly. At the same time, genes related to chlorophyll synthesis showed a nonlinear downward trend, while genes related to chlorophyll degradation showed a nonlinear upward trend. Therefore, we speculate that abnormal chlorophyll metabolism in chlorotic leaves is more likely to be related to blocking the chlorophyll synthesis pathway. In the chlorophyll synthesis pathway, the significant decrease in UROD activity and the decrease in *HEME2* (a related regulatory gene) jointly hinder the transformation from Urogen III to Coprogen III, which may be an important reason for the blocking of the chlorophyll synthesis pathway [35,36]. The expression level of the stay-green gene (*SGR*) increased significantly in chlorotic leaves, which may be related to the irregular upward trend in *PPH* and *PAO* gene expression [37]. SGR is closely related to chlorophyll degradation and may promote chlorophyll degradation by activating multiple chlorophyll degrading enzymes and the phototrapping complex. The relative expression level of the SGR gene decreased suddenly in severely chlorotic leaves, and was only higher than that in normal leaves, indicating that the stay-green gene (SGR) was involved in the process of the chlorosis of *Harumi* leaves, and chlorosis promoted the degradation of chlorophyll in *Harumi* leaves to a certain extent, but excessive chlorosis would lead to a reduced expression level of the SGR gene, and the reasons for this need to be further explored.

Previous studies have shown that the relative relationship between Chlase activity and *CAO* gene expression is closely related to the transformation from chlorophyll a to chlorophyll b [38]. In this experiment, compared with normal leaves, Chlase enzyme activity in chlorotic leaves did not show an obvious upward trend, while CAO gene expression showed a certain downward trend. This result also confirmed previous speculation to some extent that the chlorosis *Harumi* plant was more likely to be caused by abnormal chlorophyll anabolism (Figure 7).

In order to maintain normal growth, plants often remove excess reactive oxygen species through the antioxidant enzyme system [39]. Plants will produce a large number of reactive oxygen species in the process of their own metabolism or under external stress. If they are not removed in time, they will have a serious toxic effect on plant growth and development, and affect plant signal transduction and cell membrane stability [40]. It has been reported that oxidative stress is induced to cause a dramatic decrease in photosynthetic pigments [33]. In this experiment, except for CAT, the activity of other antioxidant enzymes in chlorotic leaves showed a downward trend compared to those in normal leaves. POD and APX activity decreased significantly as chlorosis increased. Therefore, it is speculated that a decrease in POD and APX enzyme activities leads to a decrease in the antioxidant capacity of *Harumi* leaves, which may lead to the content of active oxygen species exceeding the normal level and reactive oxygen species destroying chlorophyll and affecting the formation of photosynthetic pigments, resulting in metabolic disorder and chlorosis of the leaves. Further measurements of oxygen reactive species concentration are needed to confirm this hypothesis. CAT activity did not change significantly in normal, mild and moderate chlorosis leaves, but increased significantly in severe chlorosis leaves. It may be that the membrane lipid peroxidation of heavily chlorotic leaves promoted the increase in CAT activity via negative feedback regulation [41].

## 4. Materials and Methods

### 4.1. Plant Material and Treatment

According to the classification standard of Xing [42], leaves were divided into normal, mild, moderate and severe chlorotic leaves (Figure 1A). We selected nine *Citrus* trees (*Citrus reticulata* × *Citrus sinensis Harumi*) that had been grafted for five years in a small plot with strong growth and were basically the same tree shape: three trees had one treatment, with three independent real repetitions. Leaves were sampled from the farm of Danling County, Meishan City, Sichuan Province (30°02′ N, 103°52′ E, altitude 527 m), where the soil organic matter content was 4.12%, and the pH was slightly acidic (pH = 6.6). We collected the samples at 9:00–12:00 in early April at a temperature of 18–22 °C and irradiance of 11,300–11,450 kj/m^2^. For each tree, 8 leaves with varying degrees of chlorosis and 8 normal leaves, which were from autumn shoots of the previous year, were collected from 4 directions in the east, south, west and north.

The collected leaves were grouped, placed in a foam box with dry ice and brought back to the laboratory. We performed the following treatments on all samples: wash tissue surface dirt with clean water and distilled water, wipe dry with paper towels, cut the leaves avoiding the main vein portion and mix thoroughly. Some leaves were used for the determination of photosynthetic pigments (chlorophyll *a*, chlorophyll *b* and carotenoid), and others were treated with liquid nitrogen and stored at −80 °C for the determination of antioxidant enzyme activity, chlorophyll precursor substance synthesis content, intermediate metabolites of chlorophyll degradation, enzyme activities of chlorophyll metabolism and key regulatory genes.

### 4.2. Determination of Photosynthetic Parameters 

A total of 5–8 typical leaves with the same growth and similar illumination at the same leaf position for each chlorosis level (normal, mild, moderate and severe) between 9:30 and 11:00 on a clear day, avoiding the main vein, were selected and used for the determination of the photosynthetic index. Pn, Tr, Gs and Ci were monitored using a portable photosynthesis measuring instrument (LI-6400; Licor, Lincoln, NE, USA) (setting parameters: illumination 800 µmol·m^−2^·s^−1^, CO_2_ concentration 400 µmol·mol^−1^, temperature 25 °C, relative humidity 82 ± 0.5%).

### 4.3. Determination of Photosynthetic Pigments

The determination of photosynthetic pigment content was adopted and improved from the method of Arnon [43]. Fresh leaves were cut and mixed, 0.1 g was weighed and placed in a 10 mL centrifuge tube, 10 mL of 95% ethanol was added and extracted in the dark until the leaves were completely white (about 24 h), 95% ethanol was the blank control and OD values of 663 nm, 645 nm and 470 nm were measured using an enzyme calibration system (Thermo Fisher Scientific, multiskan go). We calculated chlorophyll *a* (Chl) (*a*), chlorophyll *b* (Chl *b*), carotenoid and total chlorophyll (T-Chl) content, respectively.

We calculated the concentration of each pigment in the leaves (mg·g^−1^) according to the following equations:Chl *a* content = 12.21 × OD_663_ − 2.81 × OD_645_.(1)
Chl *b* content = 20.13 × OD_645_ − 5.03 × OD_663_.(2)
Caroteniod content = (1000 × OD_474_ − 3.27a − 104b)/229 (a and b indicate the content of Chl *a* and Chl *b*, respectively).(3)
T-Chl content = Chl *a* + Chl *b*.(4)

### 4.4. Determination of Chlorophyll Synthesis Precursor Substance Content

Plant δ-5-aminolevulinic acid (δ-ALA) ELISA Kit (catalog number: ZK-8135) and Plant Bilinogen (PBG) ELISA Kit (catalog number: ZK-7842) (Shanghai Zhen Ke Biological Technology Co., Ltd., Shanghai, China) were used to determine the content of ALA and PBG. 

The method of measuring Urogen III and Coprogen III was taken from Yu et al. [44]. A 0.3 g leaf sample was weighed and the appropriate amount of liquid nitrogen was added and fully grinded, transferred to a centrifuge tube with 3 mL 0.067 M phosphate buffer (pH 6.8) and centrifuged at 12,000 rpm for 10 min; 1.5 mL of supernatant was taken and 75 μL of 1% Na_2_S_2_O_3_ was added, shaken vigorously, irradiated with strong illumination for 20 min, had its pH adjusted to 3.5 using glacial acetic acid and then was extracted with 3 mL of ether. The OD value of the water phase at 405.5 nm was measured after stratification. The content of Urogen III was calculated using the molar extinction coefficient (5.48 × 10^5^ mol^−1^·cm^−1^) of 405.5 nm. Then, the ether extract above was extracted using 1 mL 0.1 M HCl, and the salt phase OD value was determined at a wavelength of 399.5 nm. The Coprogen III content was calculated at 4.89 × 10^5^ mol^−1^·cm^−1^ (molar extinction coefficient 399.5 nm) with 3 biological repetitions. 

The determination of Proto-IX, Mg-Proto IX and Pchlide content was performed as described by Liu et al. [45]. A 0.3 g leaf sample was weighed, appropriate liquid nitrogen was added to it and it was grinded; 10 mL extraction solutions were added (acetone/ammonia water = 9:1), homogenized thoroughly and then centrifuged at 12,000 rpm for 10 min. The OD of the supernatant was measured at 575 nm, 590 nm and 628 nm, respectively, with 3 biological repetitions. The content was calculated by the following equations:Proto-IX content = 0.18016 × OD_575_ − 0.04036 × OD_628_ − 0.04515 × OD_590_.(5)
Mg-Proto IX content = 0.06077 × OD_590_ − 0.01937 × OD_575_ − 0.003423 × OD_628_.(6)
Pchlide content = 0.03563 × OD_628_ + 0.007225 × OD_590_ − 0.02955× OD_575_.(7)

### 4.5. Determination of Antioxidant Enzyme Activity

SOD activity was measured as described by Heath et al. [46]. The guaiacol colorimetric method [47] was used to determine POD activity. The determination of CAT and APX activities was determined using ultraviolet spectrophotometry [48].

### 4.6. Determination of Enzyme Activity Related to Chlorophyll Synthesis

Plant Glu-tRNAs ELISA Kit (catalog number: ZK-8377), plant Magnesium protoporphyrin IX methyltransferase (ChlM) ELISA Kit (catalog number: ZK-8381) and plant uroporphyrinogen decarboxylase (UROD) ELISA Kit (catalog number: ZK-8383) (Shanghai Zhen Ke Biological Technology Co., Ltd., Shanghai, China) were used to extract enzymes and determine GluTR, UROD and ChlM activities.

### 4.7. Determination of the Chlase (Chlorophyllase) and Mg-Dechelatase Activity

#### 4.7.1. Enzyme Extraction

The method described by Costa [49] was used for the extraction of Chlase and Mg-dechelatase. We took 1 g frozen leaves, grinded them into fine powder in liquid nitrogen and poured them into 3 mL of the following extraction buffer: 0.1 M sodium phosphate buffer (pH 6.0), 0.2% (*v/v*) Triton X-100, 30 g/L PVP, 1 mM phenyl methyl sulfonyl fluoride (PMSF) and 5 mM cysteine. The mixture was placed on a low temperature shaker at 4 °C for 1 h and then centrifuged at 9000× *g* for 20 min at 4 °C. The supernatant was crude enzyme extract.

#### 4.7.2. Preparation of Substrates

The preparation of chlorophyll refers to the method of Harpaz-Saad [35], with a few modifications. To 10 g fresh leaves of *Harumi*, 20 mL of chilled (−20 °C) 80% acetone was added and extracted in the dark at 4 °C for 12 h. It was centrifuged at 10,000× *g* for 15 min at 4 °C, and then the supernatant was taken to determine its absorbance at 645 nm and 663 nm. Chl *a* (µg/mL) = 12.7A_663_–2.69A_645_; then, we diluted the supernatant to 60 μg/mL.

#### 4.7.3. Chlorophyllase Activity

Chlase activity was measured via the method described by Suzuki et al. [50]. The reaction mixture contained 0.3 mL crude enzyme extract, 1 mL of phosphate buffer 0.1 M (pH 7.0) with 0.15% Triton X-100 and 0.3 mL acetone solution. We incubated the mixture at 40 °C for 60 min in the dark and stirred it frequently. After 60 min, 3 mL of acetone at 4 °C was added immediately to terminate the reaction, and 3 mL of hexane was added to extract the remaining Chl. The mixture was vigorously stirred until an emulsion formed, and then it was centrifuged at 9000× *g* for 2 min at 4 °C. Absorption was measured at 667 nm of the lower water layer to determine the chlase activity. The extinction coefficient was 76.79 mmol cm^−1^ [51].

#### 4.7.4. Mg-Dechelatase Activity

Substrates were prepared from Chl as described by Suzuki et al. [50] and stored in the dark at −20 °C for standby. The Mg-dechelatase activity determination procedure was taken from Suzuki et al. [35]. The reaction system consisted of 50 mM Tris-Tricine buffers (pH 8.8), 10 μL substrate and 200 μL crude enzyme extract, with a total volume of 3 mL. We incubated the mixture at 37 °C for 30 min and determined the absorbance at 692 nm. An increase of 0.001 within 30 min was expressed as a unit of enzyme activity.

#### 4.7.5. Detection of Related Gene Expression in Chlorophyll Metabolism Pathway

An RNAprep Pure Polysaccharide Polyphenol Plant Total RNA Extraction Kit (Tiangen Biotech, Beijing, China) was used to extract total RNA from the leaves, and the ReverTra Ace^®^ qPCR RT Master Mix Kit was used to synthesize cDNA. We used qRT-PCR primers (Table 2), as described by Pillitteri et al. [52]. Primers were synthesized by Tsingke Biotechnology Co., Ltd. (Beijing, China). RT-qPCR was performed using a 2X M5 HiPer SYBR Premix EsTaq (Mei5 Biotechnology Co., Ltd.) and a CFX96 Real-Time PCR Detection System (Bio-Rad, Hercules, CA, USA) instrument. 

First, the relevant gene sequence was found in *Arabidopsis thaliana*, and then the gene sequence was blasted in the Citrus Pan-genome to Breeding Database (http://citrus.hzau.edu.cn/index.php, accessed on 1 January 2023); then, primers were designed using Primer3 (http://bioinfo.ut.ee/primer3–0.4.0/, accessed on 12 January 2023) and synthesized by Sangon Biotech. The genes in the table are *HEMA1*, *HEME2*, *HEMG1*, *CHLH*, *CAO*, *CLH*, *PPH*, *PAO*, *SGR* and *Actin*. Primer 0.8 μL, ddH_2_O 7.6 μL, 2× M5 HiPer SYBR Premix EsTaq 10 μL and cDNA 1.6 μL formed the composition of the 20 μL reaction system. The quantitative PCR fluorescence reaction program was 95 °C predenaturation for 30 s, 95 °C denaturation for 5 s and 53 °C annealing for 30 s, and fluorescence signals were collected at 53 °C for 40 PCR amplification cycles. The internal reference gene was *Actin* (*Citrus* sinensis *actin*-7), and there were three technical replicates and three biological replicates for each sample. The 2^−ΔΔCT^ method [53] was used to calculate the relative expression of the gene.

### 4.8. Correlation and Statistical Analysis

The data were analyzed using Duncan’s multiple range test with SPSS 26.0 at the *p* < 0.05 level of significance. Pearson correlation analysis was used to analyze the correlation between 32 indicators of photosynthesis and chlorophyll degradation. Graphical representation was drawn using Origin 2021.

## 5. Conclusions

This experiment investigated the differences in photosynthesis characteristics and chlorophyll metabolism in the leaves of *Citrus* cultivar (*Harumi*) with different degrees of chlorosis. The results showed that photosynthesis characteristics, chlorophyll metabolism and the leaf antioxidant enzyme system were correlated. Abnormal UROD activity and *HEME2* gene expression played key roles in the chlorophyll synthesis pathway, which hindered the key pathway of transformation of Urogen III into Coprogen III and led to the inhibition of chlorophyll synthesis. At the same time, the combined action of MDCase activity and *SGR* gene expression might have caused excessive chlorophyll degradation to some extent. Abnormal chlorophyll metabolism led to a decrease in chlorophyll content, which eventually led to leaf chlorosis and affected the photosynthesis process. In addition, the decrease in antioxidant enzyme activity mainly caused by POD and APX led to a decrease in plant antioxidant capacity, and negatively promoted an increase in CAT content in severely chlorotic leaves, which led to leaf chlorosis and photosynthesis disorder.

In summary, the blocked transformation of Urogen III to Coprogen III, excessive chlorophyll degradation and the weakening of antioxidant enzyme system activity might have particularly important impacts on the chlorosis of *Harumi* leaves. Although the specific regulatory mechanism needs further study, we identified the key substances, enzymes and genes that may cause chlorosis, as well as possible abnormal chains. These data provided a basis for further follow-up studies, for example on the reason why the transformation from Urogen III to Coprogen III was blocked, methods to solve the chlorosis of *Harumi* leaves, the chlorosis mechanism of other citrus varieties and so on.

## Figures and Tables

**Figure 1 ijms-24-08394-f001:**
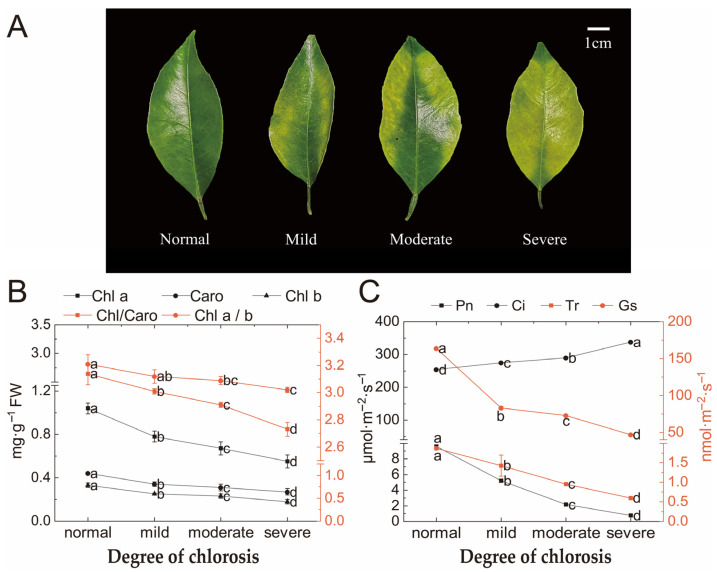
Leaves were in different degrees of chlorosis (**A**) and the effect of different degrees of chlorosis on photosynthetic pigments (**B**) and photosynthetic parameters (**C**). Values are shown as mean ± SD (*n* = 3). Different letters above the bar chart indicate significant differences between different treatments (*p* < 0.05). FW, fresh weight. SD, standard deviation.

**Figure 2 ijms-24-08394-f002:**
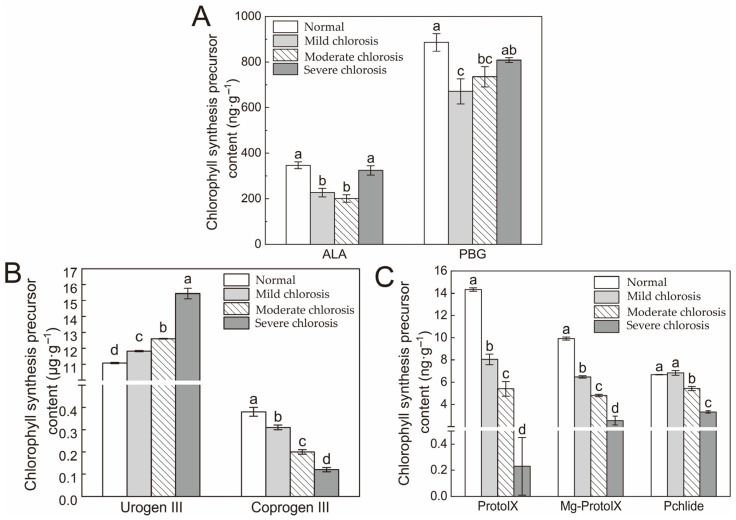
Effect of different degrees of chlorosis on the precursor of chlorophyll synthesis. ALA and PBG (**A**), Urogen III and coprogen III (**B**), Proto IX and Mg-Proto IX (**C**). Values are shown as mean ± SD (*n* = 3). Different letters above the bar chart indicate significant differences between different treatments (*p* < 0.05).

**Figure 3 ijms-24-08394-f003:**
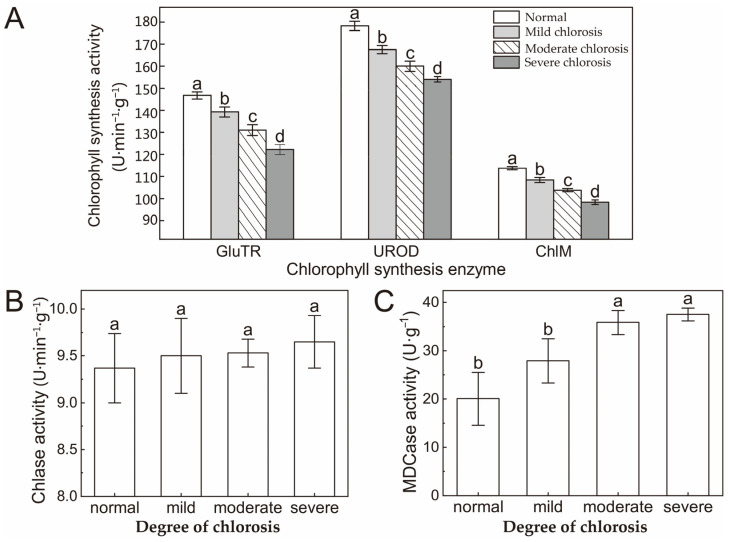
Effect of different degrees of chlorosis on GluTR, UROD and ChlM activity (**A**), chlorophyllase activity (**B**) and Mg-dechelatase activity (**C**). Values are shown as mean ± SD (*n* = 3). Different letters above the bar chart indicate significant differences between different treatments (*p* < 0.05). Chlase—chlorophyllase. MDCase—Mg-dechelatase. SD—standard deviation.

**Figure 4 ijms-24-08394-f004:**
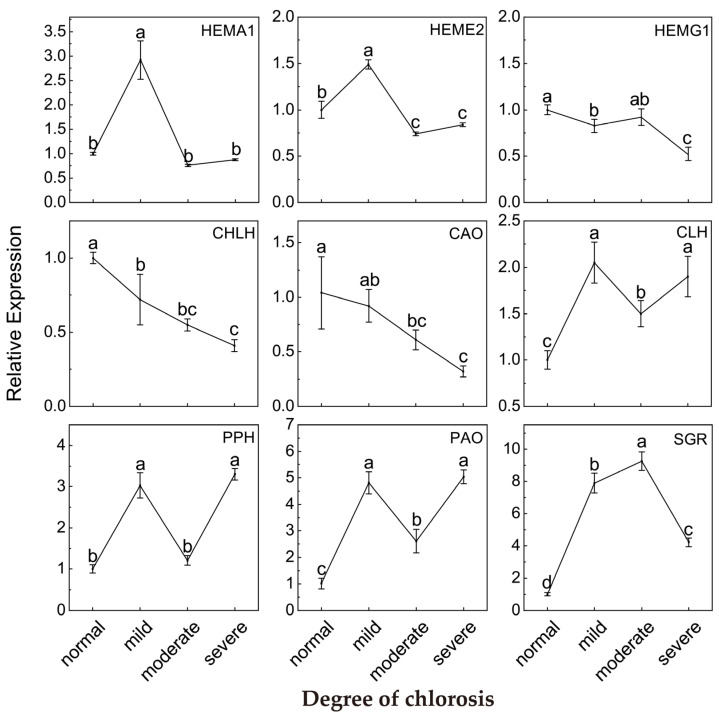
Effect of different degrees of chlorosis on the expression of *HEMA1*, *HEME2*, *HEMG1*, *CHLH*, *CAO*, *CLH*, *PPH*, *PAO* and *SGR*. Values are shown as mean ± SD (*n* = 3). Different letters above the bar chart indicate significant differences between different treatments (*p* < 0.05). SD, standard deviation.

**Figure 5 ijms-24-08394-f005:**
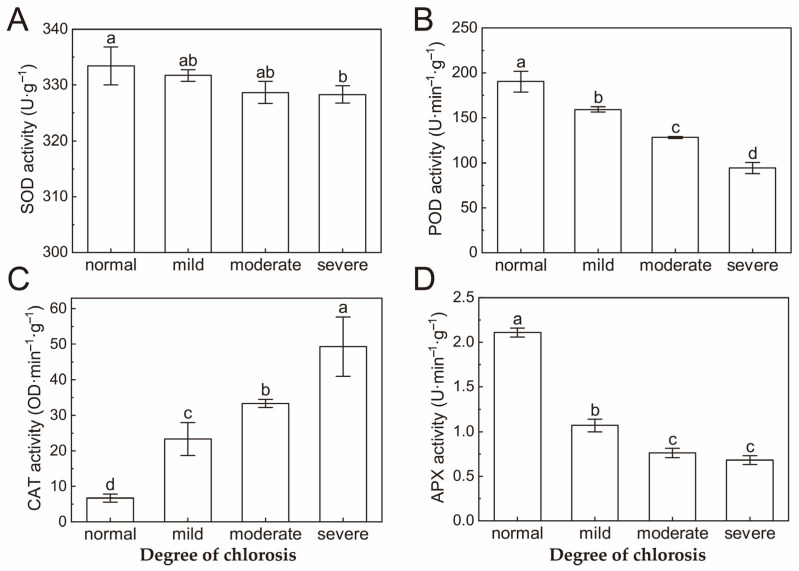
Effect of different degrees of chlorosis on SOD activity (**A**), POD activity (**B**), CAT activity (**C**) and APX activity (**D**). Values are shown as mean ± SD (*n* = 3). Different letters above the bar chart indicate significant differences between different treatments (*p* < 0.05). SD, standard deviation.

**Figure 6 ijms-24-08394-f006:**
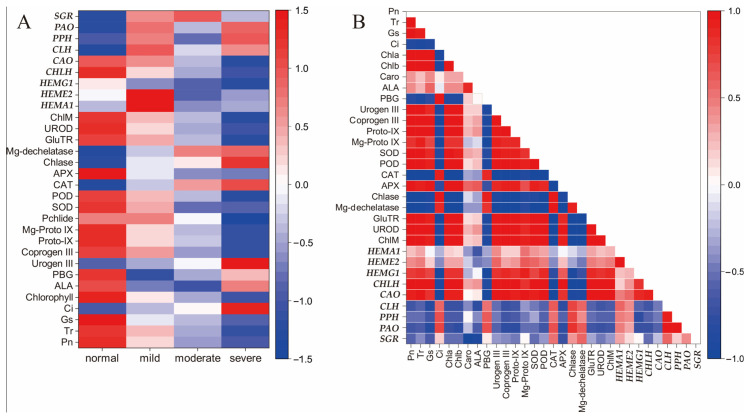
Correlation between indicators and different degrees of leaf chlorosis (**A**), and correlations between photosynthesis and chlorophyll metabolism indicators in leaf chlorosis (**B**). The positive correlation between indicators is indicated by red, and the negative correlation between indicators is indicated by blue.

**Figure 7 ijms-24-08394-f007:**
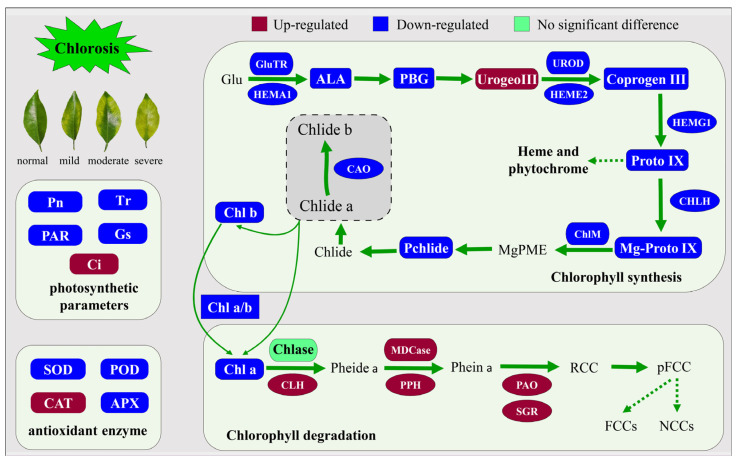
Leaf chlorosis regulatory network for chlorophyll synthesis and degradation, photosynthetic gas exchange parameters and antioxidant enzyme system in *Harumi*.

**Table 1 ijms-24-08394-t001:** Determination coefficients between chlorophyll pigments and photosynthesis parameters.

Correlation Coefficients	Pn	Tr	Gs	Ci
Chl *a*	0.9898	0.9681	0.9456	0.8462
Chl *b*	0.9806	0.9557	0.9563	0.8479
Carotenoid	0.9800	0.9258	0.9806	0.7800
T-Chl	0.9856	0.9586	0.9558	0.8402

**Table 2 ijms-24-08394-t002:** Primer sequences used for gene expression analysis.

Gene Name	Accession Number	Length	Forward (5′–3′)	Reverse (5′–3′)
*HEMA1*	XM_006472322	136	GTCTTCACCAGCACAGCATCTGA	CACAAGAGCCCACATTACGAGGAA
*HEME2*	XM_006486147	62	CTGTAGCGGAACCGAAAAATG	TCCTCGAACAGCTTTCAGCAA
*HEMG1*	XM_006444561	162	TTCTGTAGATGCTGCCGGTG	AATGTTTCCACCCCTTGGCT
*CHLH*	XM_006489925	89	GTGGCGACCCTATCAGGAAC	TGCTGCTGTGGTGGGAATAG
*CAO*	XM_006472723	57	TCGCATCCAATGCCCATAT	TTCTCGCATTTCCCATCTGTT
*CLH*	XM_006443932	62	GAAACGAATCGAGGGATCCA	CTTCAGAAACGCCACCACAA
*PPH*	XM_006482877	61	GATGCAGGTAGTTTCCCAAAAGA	GCAAGCCCGGAATTAAAACC
*PAO*	XM_006487933	60	AAGCAAGAATTTGTCTCCACTACGA	TCTGATGCTGCAGGGTCTGA
*SGR*	XM_006477286	167	CAAGGTCATCTCATCAAGGA	GATTCTACTCCGTTCTTACAAG
*Actin*	XM_006464503	195	CATCCCTCAGCACCTTCC	CCAACCTTAGCACTTCTCC

## Data Availability

Not applicable.

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
