# Peer review of "Identification of Photosynthesis Characteristics and Chlorophyll Metabolism in Leaves of Citrus Cultivar (Harumi) with Varying Degrees of Chlorosis"

_ijms, 2023, doi:10.3390/ijms24098394_

Round 1
Reviewer 1 Report
Dear Authors,
the work is interesting however the a few correction is needed. The Introduction need some references reagarding to etiolation phenomena and its reasons.
In the disscusion should be included effects of studied factors.

Author Response
Dear Editors and Reviewers:
On behalf of all the contributing authors, I would like to express our sincere appreciations of your letter and reviewers’ constructive comments concerning our manuscript entitled “Identification of photosynthesis characteristic and chlorophyll metabolism in leaves of Citrus Cultivar (Harumi) leaves with varying degrees of etiolation” (ID: ijms-2285394). Those comments are all valuable and helpful for improving our paper. According to the editor and reviewers’ comments, we have made extensive modifications to our manuscript and believe that we have fully addressed their questions and concerns. We have provided a copy of the reviewer’s comments with our point-to-point responses, and tried our best to improve the manuscript. These changes will not influence the content and framework of the paper.
We appreciate for Editors and Reviewers’ warm work earnestly, and hope that the correction will meet with approval.
Best regards,
Dr. Bo Xiong
Sichuan Agricultural University, China
Detailed responses to reviewer
Comment 1-1:
the work is interesting however the a few correction is needed. The Introduction need some references reagarding to etiolation phenomena and its reasons.In the disscusion should be included effects of studied factors.
Response 1-1:
We sincerely thank the reviewer for careful reading. According to your suggestion, we have included some recent references in the introduction to explain the phenomenon and its causes. In the discussion, we have added the effects of studied factors.
Reviewer 2 Report
The presentation of abbreviation 'Chl a' should be consistent throughout the entire manuscript (no italic in Table 1 ).
Serial number should be provided for all equations.
Author Response
Dear Editors and Reviewers:
On behalf of all the contributing authors, I would like to express our sincere appreciations of your letter and reviewers’ constructive comments concerning our manuscript entitled “Identification of photosynthesis characteristic and chlorophyll metabolism in leaves of Citrus Cultivar (Harumi) leaves with varying degrees of etiolation” (ID: ijms-2285394). Those comments are all valuable and helpful for improving our paper. According to the editor and reviewers’ comments, we have made extensive modifications to our manuscript and believe that we have fully addressed their questions and concerns. We have provided a copy of the reviewer’s comments with our point-to-point responses, and tried our best to improve the manuscript. These changes will not influence the content and framework of the paper.
We appreciate for Editors and Reviewers’ warm work earnestly, and hope that the correction will meet with approval.
Best regards,
Dr. Bo Xiong
Sichuan Agricultural University, China
Detailed responses to reviewer
Comment 2-1:
The presentation of abbreviation 'Chl a' should be consistent throughout the entire manuscript (no italic in Table 1 ).
Response 2-1:
We are so sorry for the carelessness. Based on your comments, we have made the corrections to make the word harmonized within the whole manuscript. Thanks for your correction.
Comment 2-2:
Serial number should be provided for all equations.
Response 2-2:
Thank you for reviewer’s carefully attention. For your suggestion, we have provided serial number for all equations.
Reviewer 3 Report
The authors analyzed multiple enzymes, metabolites and genes putatively implied in the etiolation process in leaves of Citrus cultivar affected by different yellowing degrees. The authors concluded that etiolation would be a consequence of alterations in the oxidative metabolism. The authors performed a valuable comprehensive study with an adequate level of replication. Unfortunately, oxygen reactive species were not measured which would have been a strong support for the conclusion. Regardless of this, I think that the title is confusing as normally etiolation is defined as prolonged growth in light absence resulting in development of etioplats in the tissue. Obviously, the process researched in the work does not match this definition. Consequently, I will suggest using a less confounding term in the title as well as providing a precise definition of the process being researched in the Introduction. Additionally, I miss a good description of the growth conditions of the Citrus trees researched and a Discussion about the environmental factors that would trigger the process at the natural ambient. I have some doubts about the statistical tests used and several sentences require to be rewritten as they are confusing and/or excessively vague.
Based on these comments, I think that the manuscript requires revision before being useful for publication.
Specific comments
Line 41. Check the usage of “supply” in this context.
Line 44. The sentence commencing with “When Chl content changes ...” should be revised. I guess that changes in Chl content would be also related with natural process (photoacclimation).
Line 49. The expression “chlorophyll a is essential for the photochemistry” sounds excessively vague. It should be rewritten.
Line 67. The sentence commencing with “Antagonistic synthesis and catabolic activities ...” is confusing. The second part of the sentence appears to contradict the first part.
Lines 68-70. Do you hypothesize that the process researched has a different cause? The interest of the manuscript will increase if a clear hypothesis is formulated.
Line 78. What does it mean with “substance”?
Lines 80-82. I understand that studying the cause of etiolation is researching the environmental conditions dealing with this process. In my opinion, what it is analysed in this study is the underlying physiological mechanism. Furthermore, a definition of “abnormal physiological etiolation” is necessary.
Lines 94-95. I cannot understand why the authors used PAR as a physiological variable. From what I understand, PAR is the incident irradiance that obviously depends on the weather conditions, daily cycle, season, location of the leaf, etc. I do not know why PAR decreased with the detiolation degree. Does it mean that gas exchange was measured under different PAR conditions depending on the detiolation degree? If this is the case, I am not sure that the results of gas exchange are comparable though the different detiolation degrees.
Line 97. Two decimal places are enough for the determination coefficient.
Line 104. I think that the Table shows determination coefficients (R2) instead of regression coefficients. Please, explain how these correlations were performed. It appears that the mean values for each etiolation degree were compared (i.e. n=4). However, if the parallel measurements of the different parameters were performed on the same sample (i.e. using the same leaf), it would be more suitable doing correlations with all pairwise data.
Lines 149-150. The sentence commencing with “Relative expression data...” should be rewritten (the fact that the expression of these genes changed in etiolated leaves does not imply or demonstrate that etiolation is due to changes in the expression of these genes).
Line 188. It is confusing how these correlations with the etiolation index were performed. Previously in the manuscript, the etiolation degree was described qualitatively (correlations require quantitative data). It is unclear how an index was calculated (similar comment for panel A in Figure 6).
Line 206-208. See my previous comment regarding to PAR.
Line 211-212. It is unclear why decreasing in chlorophyll/carotenoid ratio implies that “Chl, which is a potential cellular phytotoxin”.
Line 214. It is unclear what is being hypothesized (does it mean that the prone to etiolation is a genetic feature?).
Line 228. “The change in their trend did not show any specific regularity”. This sentence is incomprehensible.
Line 255-258. Without measurements of oxygen reactive species concentration, this hypothesis is speculative.
Lines 264-281. The environmental variables characterizing the conditions during the sample collecting should be described (temperature, irradiance, soil mixture). Date of sampling must be also indicated.
Lines 283-284. These illumination conditions should be described.
Lines 389-392. I guess that ANOVA was used to determine the statistical significance of the differences among leaf groups with different etiolation degree. Please, comment if the data fulfilled the ANOVA assumptions. I wonder if any other non-parametric test would be more appropriated for this kind of data.
Author Response
Dear Editors and Reviewers:
On behalf of all the contributing authors, I would like to express our sincere appreciations of your letter and reviewers’ constructive comments concerning our manuscript entitled “Identification of photosynthesis characteristic and chlorophyll metabolism in leaves of Citrus Cultivar (Harumi) leaves with varying degrees of etiolation” (ID: ijms-2285394). Those comments are all valuable and helpful for improving our paper. According to the editor and reviewers’ comments, we have made extensive modifications to our manuscript and believe that we have fully addressed their questions and concerns. We have provided a copy of the reviewer’s comments with our point-to-point responses, and tried our best to improve the manuscript. These changes will not influence the content and framework of the paper.
We appreciate for Editors and Reviewers’ warm work earnestly, and hope that the correction will meet with approval.
Best regards,
Dr. Bo Xiong
Detailed responses to reviewer
Major comments
Regardless of this, I think that the title is confusing as normally etiolation is defined as prolonged growth in light absence resulting in development of etioplats in the tissue. Obviously, the process researched in the work does not match this definition. Consequently, I will suggest using a less confounding term in the title as well as providing a precise definition of the process being researched in the Introduction. Additionally, I miss a good description of the growth conditions of the Citrus trees researched and a Discussion about the environmental factors that would trigger the process at the natural ambient. I have some doubts about the statistical tests used and several sentences require to be rewritten as they are confusing and/or excessively vague.
Response:
We feel great thanks for your professional review work on our article. As you are concerned, the term used to describe the leaves is really inappropriate. According to your suggestion, we have corrected “etiolation” to “chlorosis” in the manuscript. In addition, the growth conditions of the Citrus trees are supplemented, and the statistical methods used have been replied in the revised manuscript.
Specific comments
Comment 1:
Line 41. Check the usage of “supply” in this context.
Response 1:
We sincerely thank the reviewer for careful reading. We have checked the usage of “supply” and corrected the “supply” into “provision” .
Comment 2:
Line 44. The sentence commencing with “When Chl content changes ...” should be revised. I guess that changes in Chl content would be also related with natural process (photoacclimation).
Line 49. The expression “chlorophyll a is essential for the photochemistry” sounds excessively vague. It should be rewritten.
Line 67. The sentence commencing with “Antagonistic synthesis and catabolic activities ...” is confusing. The second part of the sentence appears to contradict the first part.
Comment 2:
Thanks for your great suggestion on improving the accessibility of our manuscript. We think this is an excellent suggestion. We have revised and rewritten the sentence, and added the relevant description that changes in chlorophyll content are related to natural processes.
Comment 3:
Lines 68-70. Do you hypothesize that the process researched has a different cause? The interest of the manuscript will increase if a clear hypothesis is formulated.
Response 3:
Thanks for your comments, we agree with you that a clear hypothesis can increase the interest of the manuscript. The process of chlorophyll degradation is described in detail by a flow chart in Figure 7 of the discussion section. All factors affecting its metabolic process can affect chlorophyll decomposition, including environmental and genetic factors.
Comment 4:
Line 78. What does it mean with “substance”?
Response 4:
“Substances” refer to various intermediates produced in the process of chlorophyll metabolism. We have deleted this word because of the ambiguity here, and revised this sentence.
Comment 5:
Lines 80-82. I understand that studying the cause of etiolation is researching the environmental conditions dealing with this process. In my opinion, what it is analysed in this study is the underlying physiological mechanism. Furthermore, a definition of “abnormal physiological etiolation” is necessary.
Response 5:
The purpose of our research is exactly what you think. Through previous soil tests, we found that low Mg content in soil may be one of the reasons leading to leaf chlorosis. Therefore, we mainly studied and analyzed the internal physiological mechanism of chlorotic leaves in this experiment. In addition, we added the definition of “abnormal physialization etiolation”.
Comment 6:
Lines 94-95. I cannot understand why the authors used PAR as a physiological variable. From what I understand, PAR is the incident irradiance that obviously depends on the weather conditions, daily cycle, season, location of the leaf, etc. I do not know why PAR decreased with the detiolation degree. Does it mean that gas exchange was measured under different PAR conditions depending on the detiolation degree? If this is the case, I am not sure that the results of gas exchange are comparable though the different detiolation degrees.
Line 206-208. See my previous comment regarding to PAR.
Response 6:
We thank the reviewer for pointing out this issue. We carefully analyzed the data of PAR and the instrument used to measure the data, found that PAR data was inappropriate. PAR should be a constant quantity. We have revised these contents, and apologise for our careless mistakes.
Comment 7:
Line 97. Two decimal places are enough for the determination coefficient.
Response 7:
Thanks for your careful checks, but the values of the determination coefficients are close to each other, so two decimal places may make the values the same as each other, so we decide to use four decimal places to maintain the accuracy.
Comment 8:
Line 104. I think that the Table shows determination coefficients (R2) instead of regression coefficients. Please, explain how these correlations were performed. It appears that the mean values for each etiolation degree were compared (i.e. n=4). However, if the parallel measurements of the different parameters were performed on the same sample (i.e. using the same leaf), it would be more suitable doing correlations with all pairwise data.
Response 8:
Thank you for this insightful suggestion. We have carefully thought and calculated the correlation again and found that the use of regression coefficient is inappropriate.We have adopted your suggestion, done correlations with all pairwise data and modified the data in Table 1 as the certainty coefficient, and also modified the relevant content.
Comment 9:
Lines 149-150. The sentence commencing with “Relative expression data...” should be rewritten (the fact that the expression of these genes changed in etiolated leaves does not imply or demonstrate that etiolation is due to changes in the expression of these genes).
Response 9:
We have rewritten this part according to the Reviewer’s suggestion.
Comment 10:
Line 188. It is confusing how these correlations with the etiolation index were performed. Previously in the manuscript, the etiolation degree was described qualitatively (correlations require quantitative data). It is unclear how an index was calculated (similar comment for panel A in Figure 6).
Response 10:
We have carefully calculated the correlation again and found that the use of regression coefficient is inappropriate.We have adopted your suggestion and modified the data in Table 1 as the certainty coefficient, and also modified the relevant content.
Comment 11:
Line 211-212. It is unclear why decreasing in chlorophyll/carotenoid ratio implies that “Chl, which is a potential cellular phytotoxin”.
Response 11:
There's no correlation between “decreasing in chlorophyll/carotenoid ratio” and “Chl, which is a potential cellular phytotoxin”. “Chl, which is a potential cellular phytotoxin” is only true in certain situations, the last two sentences of the second paragraph in the discussion section illustrate the relevant situation.
Comment 12:
Line 214. It is unclear what is being hypothesized (does it mean that the prone to etiolation is a genetic feature?).
Response 12:
We speculate that genetics may be one of the causes of chlorosis in leaves of Harumi, but the specific situation needs to be further studied.
Comment 13:
Line 228. “The change in their trend did not show any specific regularity”. This sentence is incomprehensible.
Response 13:
We feel sorry for our incomprehensible expression. We have removed this sentence to avoid misunderstanding. Thanks for your correction.
Comment 14:
Line 255-258. Without measurements of oxygen reactive species concentration, this hypothesis is speculative.
Response 14:
Thank you for pointing out our inaccuracy. Our subsequent experiments will continue to determine the concentration of oxygen reactive species, and the hypothesis has been revised in this manuscript.
Comment 15:
Lines 264-281. The environmental variables characterizing the conditions during the sample collecting should be described (temperature, irradiance, soil mixture). Date of sampling must be also indicated.
Lines 283-284. These illumination conditions should be described.
Response 15:
Thanks for your suggestion. We have added the environmental conditions related to sample collection, date of sampling and illumination conditions to the materials and methods section of the manuscript.
Comment 16:
Lines 389-392. I guess that ANOVA was used to determine the statistical significance of the differences among leaf groups with different etiolation degree. Please, comment if the data fulfilled the ANOVA assumptions. I wonder if any other non-parametric test would be more appropriated for this kind of data.
Response 16:
Thanks for your comments, we agree with you about comments for ANOVE. The data fulfilled the ANOVA assumptions and we have added the relevant information to the manuscript. We don't know much about other non-parametric test which would be more appropriated for this kind of data, and we will try to find other methods for analysis later.
Reviewer 4 Report
The article is devoted to the study of leaf chlorosis of citrus plant varieties in the field.
The authors obtained interesting results of interest for modern plant biology.
Despite this, the article contains a number of inaccuracies that need to be corrected.
The authors confuse the terms chlorosis and etiolation. Etiolation is a process in flowering plants grown in the partial or complete absence of light.
Etiolation increases the likelihood that a plant will reach a light source, often from under the soil, leaf litter, or shade from competing plants.
And the authors in the article studied chlorosis, a condition in which leaves produce insufficient amounts of chlorophyll.
In connection with this, the word "etiolation" must be removed from the entire manuscript.
And what caused chlorosis was necessary to understand.
Causes of chlorosis are manifold lack of minerals in the soil (Fe, Mn, Zn, N), soil pH, damaged roots, pesticides and herbicides, the presence of pathogens.
It is necessary to understand the age of the leaves used in the experiment, the tier of the tree where they were collected.
The genes are not deciphered, it is necessary to explain specifically the genes choice.
It is necessary to clarify the age of plants, the intensity of the light flux, temperature, humidity, rainfall, soil composition, the number of treatments with herbicides and pesticides.
There are many inaccuracies and typos in the article, the text needs to be corrected.
It is necessary to clarify how the authors bind antioxidant enzymes and pigment biosynthesis; this is not clear in the manuscript.
It is necessary to specify the plant variety.
Author Response
Dear Editors and Reviewers:
On behalf of all the contributing authors, I would like to express our sincere appreciations of your letter and reviewers’ constructive comments concerning our manuscript entitled “Identification of photosynthesis characteristic and chlorophyll metabolism in leaves of Citrus Cultivar (Harumi) leaves with varying degrees of etiolation” (ID: ijms-2285394). Those comments are all valuable and helpful for improving our paper. According to the editor and reviewers’ comments, we have made extensive modifications to our manuscript and believe that we have fully addressed their questions and concerns. We have provided a copy of the reviewer’s comments with our point-to-point responses, and tried our best to improve the manuscript. These changes will not influence the content and framework of the paper.
We appreciate for Editors and Reviewers’ warm work earnestly, and hope that the correction will meet with approval.
Best regards,
Dr. Bo Xiong
Detailed responses to reviewer
Comment 1:
The authors confuse the terms chlorosis and etiolation. Etiolation is a process in flowering plants grown in the partial or complete absence of light.Etiolation increases the likelihood that a plant will reach a light source, often from under the soil, leaf litter, or shade from competing plants.And the authors in the article studied chlorosis, a condition in which leaves produce insufficient amounts of chlorophyll.In connection with this, the word "etiolation" must be removed from the entire manuscript.
Response 1:
Thanks for your great suggestion on improving the accessibility of our manuscript. We have corrected the “etiolation” into “chlorosis” in the entire manuscript, and modified the relevant expression.
Comment 2:
And what caused chlorosis was necessary to understand. Causes of chlorosis are manifold lack of minerals in the soil (Fe, Mn, Zn, N), soil pH, damaged roots, pesticides and herbicides, the presence of pathogens.
Response 2:
We think this is an excellent suggestion. We have added a few sentences to introduce the reasons for chlorosis in introduction and discussion.
Comment 3:
It is necessary to understand the age of the leaves used in the experiment, the tier of the tree where they were collected.
It is necessary to clarify the age of plants, the intensity of the light flux, temperature, humidity, rainfall, soil composition, the number of treatments with herbicides and pesticides.
Response 3:
Thanks for your suggestion. We have added the age of the leaves and plants, the environmental conditions related to sample, date of sampling and other to the materials and methods section of the manuscript.
Comment 4:
The genes are not deciphered, it is necessary to explain specifically the genes choice.
Response 4:
Thank you for this insightful suggestion. First, the relevant gene sequence was found in Arabidopsis thaliana, and then the gene sequence was blasted in Citrus Pan genome to Breeding Database (http://citrus.hzau.edu.cn/index.php), and then primers were designed by peimer and used as preferred. We have added this selection method to Section 4.7.5 of the manuscript.
Comment 5:
There are many inaccuracies and typos in the article, the text needs to be corrected.
Response 5:
We were really sorry for our careless mistakes. Thank you for your reminder. We tried our best to improve the manuscript and made some changes to the manuscript. These changes will not influence the content and framework of the paper.
Comment 6:
It is necessary to clarify how the authors bind antioxidant enzymes and pigment biosynthesis; this is not clear in the manuscript.
Response 6:
Thanks for your suggestion, we have added a discussion on the relationship between antioxidant enzymes and pigment synthesis in the discussion section
Comment 7:
It is necessary to specify the plant variety.
Response 7:
Thanks to your reminder, the plant variety used in this manuscript has already been specified, Harumi, which is a late-maturing hybrid citrus [‘F 2432’ Ponkan (Citrus reticulata) × Kiyomi tangor (Citrus unshiu × Citrus sinensis)]
Reviewer 5 Report
The paper by Xiong et al, studied the effect of etiolation level on chlorophyll metabolism and photosynthesis characteristics. In a first reading, the paper is well written. It has a correct introduction and extensive material and methods. Regarding the material and methods we can conclude that it fulfills the objectives of the study and is exhasutive.
When we analyze the results, the presentation of the results is adequate although there are some errors that need to be rectified.
In the last lines of the first paragraph of section 2.1, the authors discuss the result, and obviously, this part should be included in the discussion, and not in the results. We refer to the sentence "These results indicated that leaf etiolatio...".
In section 2.2. (Chlorophyll synthesis precurcursors), the authors comment "Pchlide content first increased and then decreased.... (Figure 2C)". If we look at Figure 2C, this is not the result obtained; in fact there is no increase between normal and mil etiolation, and then we observe that there is a decrease. There must be significant differences to be able to corroborate that there is a decrease or an increase. In the last sentence of this section, the authors discuss the results again.
With respect to these last lines, we must first pass them to the discussion. Furthermore, the conclusion reached by the authors is not correct. If we look at Figure 2B, at the same time that Urogen III increases with the degree of etiolation, Coprogen III biosynthesis decreases. Therefore, based on Figure 7, would it not be logical to think that it is the other way around, i.e., that Urogen III accumulates because this precursor is not transformed into Coprogen III?
In section 2.3 Activity chlorophyll synthesys and degradation enzyme, the authors comment that Chlase activitiy was lower in normal leaves. This is not correct, since if we look at Figure 3B, there are no significant differences in this activity taking into account the level of etiolation of the leaves. Likewise, in MDCase activity, although there is a tendency to increase, we see that there are only differences between normal and mild etiolation leaves, and between moderate and severe etiolation. Therefore, this section needs to be rewritten.
In section 2.5 Antioxidant enzyme activity, in the last lines of the paragraph the previous mistakes are made again; the authors comment that there are no differences in APX activity between moderate and severe etiolation, however, when observing figure 5D this is not true, since the graph shows differences between all levels of etiolation. In the same paragraph, it is commented that all enzyme activities were inhibited by etiolation compared to normal leaves. Regarding this statement, how do the authors know that this is an inhibition? This should be discussed in the discussion section, and possibly it would be more plausible to think that rather than an inhibition, it is a lack of activity associated with the senescence process and consequently a lower amount of enzyme. Either case, it should be discussed and supported at least with bibliographic references.
In the discussion section it is mentioned that etiolation increases PAR. How is this possible? PAR is an environmental variable measured by the IRGA, but it is totally independent of the leaf. Whether the leaf receives more or less PAR does not depend on the level of etiolation, but on the light source, or the orientation of the leaf when taking the measurement. Are the authors clear about how the IRGA operates? Furthermore, in an paper of 15 pages, there are less than 2 pages dedicated to the discussion, despite the fact that there are a large number of results. Moreover, in many sections it is merely descriptive, and no explanation is given as to why the results obtained; for example, why do photosynthetic parameters decrease with increasing etiolation?
Oxidative stress is also mentioned in the discussion, however, how do you know that oxidative stress exists? MDA has not been measured, chlorophyll fluorescence has not been measured. It would be more logical to think that etiolation and chlorophyll loss is associated with the senescence process, and consequently antioxidant activity would be lower due to the simple fact of recycling of all leaf compounds. In summary, the discussion is very poor in breadth and detail, and another approach beyond oxidative stress should be considered.
The article should be rewritten and resubmitted once the errors are corrected and the discussion is modified and expanded.
Author Response
Dear Editors and Reviewers:
On behalf of all the contributing authors, I would like to express our sincere appreciations of your letter and reviewers’ constructive comments concerning our manuscript entitled “Identification of photosynthesis characteristic and chlorophyll metabolism in leaves of Citrus Cultivar (Harumi) leaves with varying degrees of etiolation” (ID: ijms-2285394). Those comments are all valuable and helpful for improving our paper. According to the editor and reviewers’ comments, we have made extensive modifications to our manuscript and believe that we have fully addressed their questions and concerns. We have provided a copy of the reviewer’s comments with our point-to-point responses, and tried our best to improve the manuscript. These changes will not influence the content and framework of the paper.
We appreciate for Editors and Reviewers’ warm work earnestly, and hope that the correction will meet with approval.
Best regards,
Dr. Bo Xiong

Round 2
Reviewer 5 Report
The authors have satisfactorily corrected the suggestions made in the previous review. However, having reviewed the new version, I am still of the opinion that the discussion is short, and needs to be written in more detail. The authors have a large number of results that should be presented in more depth in the discussion section.